# Receiving Notification of Unexpected and Violent Death: A Qualitative Study of Italian Survivors

**DOI:** 10.3390/ijerph191710709

**Published:** 2022-08-28

**Authors:** Diego De Leo, Annalisa Guarino, Benedetta Congregalli, Josephine Zammarrelli, Anna Valle, Stefano Paoloni, Sabrina Cipolletta

**Affiliations:** 1Australian Institute for Suicide Research and Prevention, Griffith University, Brisbane 4122, Australia; 2Slovene Center for Suicide Research, Primorska University, 6000 Koper, Slovenia; 3De Leo Fund, 35137 Padua, Italy; 4Autonomous Syndicate of Police (SAP), 00184 Rome, Italy; 5Department of General Psychology, University of Padua, 35100 Padua, Italy

**Keywords:** traumatic death, suicide, homicide, road accident, death communication, notifiers, survivors

## Abstract

(1) Background: The loss of a significant person can be especially traumatic when death comes without warning and is due to causes such as suicide, murder and accidents. The way an individual is informed about the loss can affect the way of adapting to the loss and the quality of life of survivors. Communication modalities of the notifier may deeply influence the bereavement process. Aim: The present investigation aimed to explore the experience of those who received communication of such a type of death by a professional figure. (2) Method: Snowball sampling was used to recruit the participants to this qualitative study. Social networks, word of mouth and researchers’ acquaintances were used, including clients of the NGO De Leo Fund. A total of 52 Italian people (eight males and forty four females, mean age = 49.44; SD = 14.23) who received notification of the death of a loved one by police officers or by health professionals participated in the study. Deaths involved cases of suicide, murder, road accident and mountain accident. (3) Results: The following four key themes were identified: (a) how the communication took place; (b) reactions; (c) support; and (d) coping strategies. Only 22 participants reported having received clear information about the dynamics of what happened; the rest of the sample obtained only poor or ambiguous information. The majority of participants sought or received informal support from family and friends immediately after notification; however, some participants experienced a total lack of support. The opportunity to see the body of the loved one for the last farewell, when denied, caused discomfort in recipients of the news. (4) Conclusions: Even the impactful notification of a traumatic death such as suicide or road accident can be mitigated by the appropriate behaviour and attitude of notifiers, who should always avoid providing generic or ambiguous information about what happened. The notification process should not end with the simple transmission of the communication, but should also look at the following phases by including referral to supportive networks or bereavement services, aimed at assisting individuals in the immediate aftermath but also in the long term.

## 1. Introduction

The loss of a significant person is by its nature a shocking and traumatic event [1], especially in cases where death comes without warning [2] and is due to external and violent causes [3] as in the case of suicide, murder and accidents. Mourning for an unexpected and violent death in the literature is defined as ‘traumatic bereavement’, as the painful consequences related to the loss of a loved one add to those related to the event that was the cause of death, and can provoke the typical symptoms of traumatic experience in those who remain [4]. In these circumstances, adaptation is more difficult than in other types of loss [5], especially when adequate support is lacking [6]. Indeed, compared to bereavement by natural causes, bereavement by traumatic death can lead to a slower adaptation process and can increase the risk of mental disorders, such as post-traumatic stress disorder and depression [2,7].

A crucial aspect related to the experience of death is the communication of the same to relatives and friends by professionals: the notification of death is part of breaking bad news and represents a two-way process [8]. In the case of unexpected and violent death, the task usually involves health workers, police officers or fire-fighters (notifiers) from one side, and partners, family members and friends—known as "survivors" (recipients)—from the other [9]. For both parties, the communication of death represents an experience of strong emotional impact as it is very stressful for the notifier [10] and more so for the recipient. For the latter, from the very first moment of the bereavement process, it involves a relevant change in the narrative of the self [11]. 

The way in which an individual is informed of the death of a loved one can affect the way of adapting to the loss and the quality of life of survivors [12]. Studies have shown that an adequate and sensitive communication of death can be considered as a form of secondary prevention towards the reactions of relatives, which can help to avoid further traumatization [13,14]. In fact, the communication modalities of the notifier may deeply influence the beginning of the grieving process as they can either have a negative impact by intensifying the trauma or a positive one, by mitigating the trauma itself [11].

To facilitate the handling of this communication, notifiers can use various protocols, including GRIEV_ING—specifically built for death notification [15]—or SPIKES [16] and ABCDE [17], initially developed for communicating unfavorable prognoses, but also applicable in cases of death. Several authors underline the importance of making the communication in a private space, where what happened can be described in plain words, and notifiers can make themselves available to answer any question in an empathetic and supportive way [9,18,19,20,21,22]. According to Ahmady and colleagues [23], in fact, the factors that are most correlated with the possible consequences of loss are also conducive to the attitude of the notifier, the comprehensibility of communication, respect for privacy and the ability to answer questions. 

Studies with family members who have lost a loved one highlighted the usefulness of some specific interventions implemented by notifiers. For example, Parrish and colleagues [24] found that 71% of the sample investigated was satisfied with the level of care and emotional support received in the emergency room. In a sample of 58 relatives of 48 individuals who died in the emergency room, Jurkovich and colleagues [25] highlighted that the most important aspects during communication were respect for privacy, an attentive attitude of the notifier, and adequate information about what happened, as well as the ability to provide clear messages and answer questions. In a study by Peters et al. [26], suicide survivors felt supported by the caring attitude of the notifier who, for instance, made sure they were not alone at the time of the notification, dedicated sufficient time and asked what they needed. 

A few studies also considered types of intervention that could be useless or harmful, such as the coldness of the notifier, lack of sufficient information or excessive empathy [24,25,26,27].

Although specific protocols and suggestions have been developed to facilitate notifiers in the handling of death notification, the perception and impact of how death notification is carried out may vary among recipients. Studies on the topic of death notification do exist in the literature (e.g., [28]), but there are no studies that specifically focus on receiving the notification of the death of a loved one in unexpected and violent circumstances as experienced by recipients. The present study aimed to bridge this gap by exploring the experience of a group of Italian survivors who received the communication of such a type of death by a professional figure.

## 2. Methods

For the purpose of the study, the following factors were investigated: (a) characteristics of the setting in which the communication took place (where and by what means); (b) type of professional figure involved in the notification task; (c) verbal and non-verbal aspects of the communication; and (d) ways in which the experience of death notification were lived.

This investigation is part of a broader research project, called ‘IRIS’ exploring the aspects, reactions and feelings involved in the process of communicating an unexpected and violent death, from the perspective of both notifiers (most often police officers and health workers) and recipients (family and friends). This part of the research analyses the perspective of the recipients through a qualitative interpretative research design [29]; this was considered the most suitable for capturing the experience of the individuals and their particular perspective of the phenomenon they were involved in [30]. An online self-administered questionnaire with 12 open questions was created ad hoc (see Table 1) in order to respect the criterion of economy and reach as large a sample as possible [31]. 

Given the peculiar characteristics of the chosen population and the consequent difficulty in reaching participants, snowball sampling was used to recruit participants [32]. The network of contacts of the authors was used to find the first participants; the latter helped in finding other survivors who lived the same experience. Word of mouth and use of social networks (Facebook and Instagram) favored the process; in addition, the research project was proposed to users of the NGO De Leo Fund. In this case, those interested in taking part in the study were provided with a link for accessing the online questionnaire set up for the study. A total of 52 Italian people (8 males and 44 females, mean age = 49.44; SD = 14.23) who received the notification of the death of a loved one by the police (Carabinieri, police officers) or health professionals (ambulance workers, doctors, nurses) participated in the study. Violent and unexpected deaths involved cases of suicide, murder, road accident and mountain accident. Table 2 shows the main characteristics of the sample. Data collection lasted from January 2021 to July 2021, and, following the criteria for qualitative research, data collection and analysis of the questionnaires proceed simultaneously. Sampling ended once theoretical saturation was reached, that is the point at which gathering more data does not generate more information related to the research questions [30].

The questionnaire was conceived while keeping in mind the danger of the re-traumatisation of participants, whose risk for suicide is potentially elevated [33]. Indeed, an online questionnaire provided the participants the freedom to more easily abandon filling it out if they found it was too intrusive; they were able to close the online page without any explanation.

The Ethics Committee of the University of Padua approved the research project (No. 3878). Test administration was preceded by performance pre-tests [34] on a small number of people (including five suicide survivors) to control for the efficiency of the investigation procedure. Informed consent was obtained from all subjects involved in the study.

Since the questionnaire responses were short and poorly articulated, it was more advantageous to analyse them through a thematic content analysis (TCA) which was conducted in two steps. The first step followed the 6-phase method proposed by Braun and Clarke [35], carried out by two authors and supervised by a senior author who is a trained coder. The researchers initially read and re-read the data set until the depth and breadth of the content became familiar. Then, data generated initial codes; afterwards, codes that appeared as similar were grouped together in overarching themes, relations between codes and formed themes were identified, while some themes and codes were discarded. The results were revised to evaluate if the identified themes had sufficient supporting data and if they met internal homogeneity and external heterogeneity criteria. Finally, themes were refined and named to indicate captured topics. This process led to the creation of a codebook, which was applied to all the responses. 

The second step was the quantification of the data on the basis of frequencies observed [36]. The coding of the textual material was carried out manually with the aid of coloured markers, using a "paper and pencil" procedure. The study was performed according to the Consolidated Criteria for Reporting Qualitative Research [37].

## 3. Results

Four key themes were identified, with respective sub-themes, as reported in Table 3. The main themes were as follows: (1) how the communication took place; (2) reactions (the lived experience by survivors); (3) support obtained; and, (4) coping strategies.

### 3.1. How the Communication Took Place

The first key-theme refers to the participants’ recollection and perception of how they were informed of the death of their loved one by a professional figure. Participants described the content of the communication and the aspects that remained in their recollection. 

Many participants (*n* = 22) reported that they received clear and comprehensive information regarding the dynamics of events that led to the death of their loved one or the discovery of the body; on the contrary, other participants (*n* = 21) received only generic and hasty explanations; some were forced to request more information from the notifiers to clearly understand what happened. A mother was notified by a doctor at the hospital where her son was taken after a car accident and reported: “*I received no clear information at all. I had to ask for details and whether my son had actually died*” (P43, F, 56 years old). 

In a few cases (*n* = 4), recipients received information that was difficult to understand; for example, information that was full of medical details or *"superficial", "telegraphic", "bad"* and "*hasty"* explanations.

Death notification also included non-verbal aspects, such as looks, gestures, tone of voice and attitude that partly anticipated, replaced or strengthened the verbal content expressed. Eight participants said that before the explicit communication they understood the news through signals such as the resigned tone of the notifier or the dejected look, as exemplified in this story: 

“*A policewoman rang the doorbell, and just asked me if a boy lived in the house; she was visibly upset. […] I said that at the moment he was not at home but I didn’t know where he was […] The policewoman took the elevator and left […] Then I looked out the window of my son room which was strangely wide open, and I saw a lot of people down the street, and the ambulance. And a body on the ground covered with a green cloth. And I understood […] The policewoman did not allow me to go down to the street, and calmly convinced me to remain at home with her. To my question, ‘Is it my son that I saw? Tell me, is he my son? Is he dead?’, she didn’t say anything but she looked at me in a way that I will never forget for the rest of my life. It was a ‘yes’ that wanted to be a ‘no’, but it was a ‘yes’*”(P20, F, 43 years old)

In other experiences described by participants, there was a greater sharing of grief over what was communicated between survivors and notifiers, and communication was perceived as “*sensitive*” and “*human and empathetic”.* Some survivors (*n* = 5) gratefully reported the empathetic attitude of notifiers who were sorry to report the news. Others (*n* = 13) greatly appreciated the closeness manifested by the notifier thorough gestures of support (e.g., hand on shoulder, hugs) and verbal reassurances (e.g., ‘Your loved one did not suffer’). Other participants denounced the lack of any genuine involvement on the part of the notifier; as a result, they reported unpleasant experiences described as *"hideous", "very bad",* "*totally lacking in professionalism and basic humanity*" (P9, F, 60 years). On one hand, some participants (*n* = 14) reported having met with detached and even hasty notifiers, for example: “*I had the feeling that they wanted to take off the burden of communication as soon as possible to go home and continue with living their lives. They were unaware that our life had been forever destroyed*” (P39, F, 27 years old). On the other hand, other participants (*n* = 8) perceived the notifier’s difficulty in communicating the news and felt the embarrassment of the task, as exemplified in these words:

“*I remember that the nurse who called on the phone was in great difficulty. Her voice was shaky, and this had a negative impact on me. It seemed that she was there for a sense of duty, rather than for any closeness. […]. Words were communicated in a ritual manner. I did not feel any closeness. It seemed like a mechanical task*” (P33, F, 27 years old). 

Finally, four participants reported on the rude ways in which police officers approached them, causing their sense of respect towards police to completely vanish:

*"The roughness and extreme rudeness in oral communication were really unbearable (‘Move from here’, ‘Don’t break’, ‘We can’t tell nothing’, ‘Leave the room’, ‘Do not hinder our work’, ‘You are annoying’, etc.)*” (P9, F, 60 years old).

### 3.2. Reactions (Lived Experience)

The second key-theme refers to the emotional experience of recipients when they received the notification, and the events that immediately followed.

At the time of the news, the predominant reaction relayed by 35 participants was deep pain, described through metaphors such as visual images of destruction, perforation and rupture: *"I felt hit by a giant and by a suffocating weight, and knocked down in my person"* (P48, F, 39 years old); “*I felt pierced by an awl*” (P14, F, 59 years old). Some participants (*n* = 15) felt despair and were unable to control their screaming and shouting. One survivor said “*My mother’s screams ripped the silence of that terrible night*” (P39, F, 27 years old). A similar number of participants (*n* = 16) felt dazed and confused; they spoke of the feeling of loss of lucidity as if the mind had shut down and prevented them from understanding what was happening. One participant reported that his vision became blurred: *"I remember looking at the doctor but the background around her was foggy. I don’t remember any detail of that scene"* (P42, F, 34 years old). The same number of survivors (*n* = 16) even stated that they felt estranged from their bodies or the surrounding reality, feeling as if they were floating and living in a non-real dimension, like in a movie: *"It was as if I could see the scene from the outside"* (P39, F, 27 years old). A few participants reported feeling stiff in their body, as if petrified. One participant said: “*I could not speak or cry or react in any way, as if the running of time had stopped*” (P19, F, 47 years old); “*I was speechless, I had no reactions*” (P28, F, 39 years old). Many participants explicitly described the moment of the death communication as impossible to forget, as an indelible mark on their lives and memories. Eight participants reported experiencing an indescribable emotion “*that just cannot be expressed in words…"* (P26, M, 56 years old). 

Another common experience was disbelief (for 16 participants). A number of survivors (*n* = 6) responded to the notifiers by trying to deny what was notified, for example by accusing the officers of making a mistake, or explicitly begging them that the news was not true: *"Initially I begged them to tell me that it wasn’t true, that my son wasn’t really dead, even though I had seen his body covered by a green cloth"*(P10, F, 52 years old). Anger characterised the stories of 10 participants who reacted towards the notifier driven by the belief that the notifiers themselves were somehow responsible for what happened.

Four participants expressed the immediate desire to die when they received the news, *"On that moment I would have wanted to disappear forever"* (P21, F, 59 years old). Some participants (*n* = 6) also reported experiencing somatic reactions such as nausea and vomiting, tremor, and tachycardia.

In the moments immediately following the tragic notification, eight participants consented to organ donation. For five participants, the dialogue with the health professionals was very unpleasant as the latter showed some insistence in obtaining the consent, which affected the sensitivities of family members, *"They put pressure on me to donate organs, as if my brother was needed for spare parts. A terrible experience"* (P49, M, 68 years). Several participants reported the need to see the body after the notification. The chance for a last farewell was well received (for 12 participants): *"I appreciated the attention and the fact that they let us to stay by the still warm body of our daughter"* (P4, F, 59 years old). 

### 3.3. Support

The third key-theme refers to the participants’ lived experience of support and concerns, help sought, received or not received after the notification.

About half of the participants (*n* = 23) reported having proactively sought formal support, 13 participants turned to professional support to process the loss, some of them (*n* = 8) sought psychological and psychotherapeutic individual assistance, while others felt the need to associate themselves with people who had gone through the same experience and therefore turned to self-help groups: “*I called a friend bereaving like me and asked her for help. I was immediately contacted by the self-help group […]. It helped me a lot*” (P2, F, 48 years old). Other survivors (*n* = 10) sought informal support within their social network to obtain comfort and concrete help, such as child management in the family, organization of the funeral, etc. 

The need for support both at the psychological and practical level was expressed by several participants: “I needed everything, and I would have needed the support of a person prepared for managing this type of trauma (health worker or other). […] I point out that perhaps this type of support should be made available to everyone and provided soon, if not immediately, after the traumatic event "(P32, M, 61 years old); "I would have needed someone from the beginning to tell me what to do: the first night alone was hell" (P34, F, 23 years old). Other participants (*n* = 23) received immediate help from their own social network without asking for it, as family and friends provided practical help, for example, by accompanying the bereaved person to the accident site, buying groceries, keeping the house tidy, etc. "We were helped by our neighbors. They were crucially important because they accompanied us to the hospital, and in the following days helped us with the funeral arrangements, because my mother and I were totally out of it" (P10, F, 52 years old). Other participants (*n* = 18) received “moral” support, as one of them states "A dear friend came to give me support; with him I was able to immediately let off steam, cry and talk. I would say that my friends helped me a lot” (P19, F, 47 years old). 

In a few cases, the notifier also played an important supportive role. Four participants indeed reported that the notifier was very gentle and humane, despite being the deliverer of such terrible news: “*If it hadn’t been for the police, I don’t know how things would have turned out…They gave me the strength to react*” (P34, F, 23 years old); “*The closeness of the nurse who informed me of the death gave me courage…*” (P40, F, 45 years old). Conversely, nine participants experienced no support at all. Four survivors felt completely alone and even marginalised by society as they did not feel any real support from the persons next to them, with the belief that no one could fully understand the pain of the loss they suffered, as can be seen in these words “*In that moment, I did not feel anyone close to me. I felt that no one could understand my pain and that of my mother*” (P45, F, 51 years old). 

Three participants were bewildered after the notification, and they did not know how to organize things, such as being accompanied to the accident site or organizing the funeral; they underlined the need to receive information from police officers, stating that “*Perhaps the police could have given us references of any support association available nearby, and also useful indications for the most appropriate type of legal support*” (P16, F, 48 years old). The latter need appears particularly critical in cases of car accidents. There were two participants who lost a loved one in a road accident who reported that they did not receive legal justice, as described in these words:

“*No communication can change the facts, true! However, what could make the situation more bearable is the correct flow of justice demonstrating a real search for the truth. This should start from the recognition of the rights of the victims, and not as the prosecutor said […] ‘The dead is dead, let’s help the alive’, which turns to be—in practice—the very one who was responsible for the disaster! Incredibly, this is what they have done! It is the shameful arrogance and superficiality of the institutions, starting with the judiciary system, which creates further victimization*” (P38, F, 80 years old).

### 3.4. Coping Strategies

The fourth key theme refers to coping strategies understood as the ways in which participants coped with the loss and adapted to the new situation.

Some participants reported adapting to the loss by finding strength in themselves (two participants) or in their own reference systems, such as family (four participants). Some other survivors coped with the loss through writing (*n* = 5), as a way to communicate with the missing person and express their feelings, as described in the following excerpt: *"This is the second time I have written these things; the first time I wrote them on a pad I use to communicate with my brother. When I feel so bad, I write to him’* (P20, F, 43 years old); other people (*n* = 3) chose to participate in research projects on the theme of bereavement, with the hope that their experience might have been of help to others living the same kind of loss. For example, some participants provided the motivations that prompted them to fill in our questionnaire. “*I filled in because I think it is important that certain things are known better. That they don’t go unheard and unknown*” (P36, F, 49 years old); “*I believe a lot in these initiatives* [as the present research], *and I hope that this contribution of mine can help you*” (P33, F, 27 years old).

After a loss due to suicide, only three participants of our sample referred to having tried hard to make sense of the suicide of their loved one. The following is such an example: “*Understanding if my brother’s extreme gesture was a meditated choice or not…Well, I don’t accept it anyway! I think it was just a moment of pure madness, and I hope he didn’t feel any pain*” (P28, F, 39 years old). Other participants tried to cope with the loss by looking for new activities, and new projects in which to invest emotionally to distract them from the past.

Finally, eight participants described their loss as a learning opportunity, something that could favour a major change in their social or work commitment, as can be seen in these words:

“*I work with fragile people every day, and I continually question myself about my role: what is the value of my words and the actions for my clients, as well as their families. How can I better understand their situation and how this can help me to help them better?*” (P48, F, 39 years old).

## 4. Discussion

This qualitative study offers an overview of the death notification experience as reported by a sample of Italian survivors who lost a loved one due to suicide (majority of participants), murder, road accident or mountain accident. This qualitative investigation could be the first specific contribution on the subject of the notification of unexpected and violent death from the perspective of the recipients. As such, it could provide useful insights into this particular angle of observation, and eventually promote better management of the difficult task of notifying on those types of death.

Not unexpectedly, many more women than men replied to the online questionnaire; a good part of them were mothers who lost a child and reported their feelings about the loss. This is in line with the literature, according to which mothers are more likely to express their feelings of pain than fathers [38,39]. According to various studies (e.g., [28,40]), the notification of death represents a painful moment of which recipients keep a very detailed memory; in fact, many of the participants in this study described with considerable emotional intensity the visual, auditory and olfactory details related to that moment. 

As reported by participants, the characteristics of the setting in which the notification occurs also play an important role. Indeed, receiving the notification in a chaotic space (e.g., hospital corridors) or via the telephone was an unpleasant experience for participants, which confirms the literature data [21,25] and suggests that these modalities should be avoided as much as possible. 

Verbal and non-verbal contents adopted by the notifier appeared of relevance; in fact, modalities of communication can contribute to a secondary traumatization [13,14]. In our sample, individuals described it to be particularly unpleasant to have received generic or unclear information. The words said can be much more important than other characteristics of the person in charge of giving the notification [21]. Therefore, according to studies [18,23,25,41], and as confirmed by the participants of this research, the use of clear and unambiguous language, devoid of technical terms, appears essential. Knowledge of death circumstances helps to conceive of the reality of death [11]. 

Many participants reported having anticipated the verbal notification of death from non-verbal aspects shown by the notifier (such as appearing sad, with a trembling tone of voice). The notifier should be aware of their own attitudes and behaviours, since these can directly influence the reactions of survivors [42]. When the notifier was empathetic, this was always appreciated, and the communication style was perceived as sensitive and appropriate, confirming the results of Janzen and colleagues [11]. Several authors have underlined the importance of an empathic approach during the notification process [18,20,22]. According to Sep and colleagues [43], this modality favours a reduction in the physiological response triggered by the bad news, which could be partly responsible for the particular vividness of the memory of that communication. 

Managing the recipient’s emotional reactions represents a difficult task for the notifier [44] and, therefore, anticipating their feelings and reactions can help in conducting the notification process in the least possible traumatic way and in the meantime laying the foundations for a healthy bereavement process [11]. The reactions observed in our sample can be traced in the two macro-categories theorised by Goodrum [45] in a study in which interviewed family members who had lost a loved one to homicide which are as follows: (1) expressions of emotional upheaval, and (2) attempts to contain it. In the first macro-category, we may find pain, despair, bodily reactions and reactions of anger towards the notifier, all frequently reported in the literature (e.g., [22,42,46,47]). Besides these feelings, our participants reported feeling traumatised by the violent and unexpected news of the death and experiencing the sensation of dying and the desire to die. 

In the second macro-category, those reactions implemented as a defense against a situation perceived as intolerable and unacceptable are included. Among these are shock, disbelief, estrangement feelings (such as derealisation and depersonalization), sense of emptiness, detachment, silence and attempts at denial [14,46,47,48,49,50]. In addition—albeit not attributable to these macro-categories—are a sense of guilt and resignation, also found in earlier studies [49,50,51].

The opportunity to see the body of the loved one for the last farewell, when denied, caused discomfort in the recipients of the news of our sample. They felt crucially important needs were unmet. Seeing the body could favour a better bereavement process [52,53,54]. However, this aspect appears as particularly complex, since many variables can interfere with the outcome (e.g., cause of death, severity of injuries and/or disfigurement of the body, survivors’ preparation for these scenarios, their character strength, presence of accompanying persons, etc.). To date, there is no unanimity in the literature on the positive or negative value of the vision of the body to address a firm recommendation for a standardised approach to this emotional need [55,56,57].

As argued by Mitchell [58], the notification process must not end with the simple transmission of the communication, but should also look at the following phases by including, for example, the referral to supportive networks or bereavement services, aimed at assisting the survivors in the immediacy of the loss but also in the long term. In the present research, we therefore investigated the moments following the death notification in order to understand the needs of the participants and highlight the crucial role that the notifier can have in coping with them. According to Kaul [20], in fact, after breaking the bad news, the notifier should favour the "mobilization of resources" by specifically indicating possible sources of support for the person (e.g., connection with family members and friends, psychological help, job placement programs, etc.) and, whenever possible, developing a plan to concretely access them. In our study, in the few stories in which the notifiers provided this type of support, participants very much appreciated notifiers’ behaviour; on the contrary, notifiers were criticised when they did not offer specific contacts for emotional or legal support. 

The majority of participants in this study sought or received informal support from family and friends immediately after notification, mainly in the form of emotional closeness but also of practical help. However, some participants reported experiences of a lack of support (e.g., not being accompanied to the accident site), which increased the difficulty of the moment. In this regard, after the notification, it would be important for the notifier not to leave the recipient with a sense of abandonment and loneliness. For example, in cases of necessary travels to the scene of the accident, it could be notifiers themselves who accompany the family member [16,19]. 

Practical assistance, aimed at facilitating the satisfaction of the most elementary needs, is particularly recommended for those who have recently experienced a traumatic loss [59]. Formal treatment would not be recommended initially, as victims are typically too upset to benefit from it. However, in our study many participants expressed the need to obtain immediate psychological support. Individuals affected by the sudden and violent death of a loved one, in fact, often show acute psychological responses [20]. Survivors’ need to be assisted by a professional could arise from the need to cope better with their difficulties [60]. A recent study by Aoun and colleagues [61], on support needs in relation to various types of loss highlighted that family members who have suddenly lost a loved one frequently report the need for immediate support, both formal and informal, as the latter may not be enough.

In the months following the death notification, many participants in our study sought formal support from professionals and peer groups to facilitate the bereavement process. The benefits of these interventions for the bereaved are well established [62,63]. The support can be understood as a resource that may facilitate the creation of new meanings; this process can be more or less structured, depending on if it is performed by psychologists, psychotherapists, self-help groups or family and friends, and can promote the integration of the loss into one’s self-narration [64]. In the recipient, in fact, the notification of death can lead to a breakdown in the narration of the self and their own ordered cosmos of meanings [11]. The process of reconstruction of meaning in the participants takes place through various strategies, such as retelling and re-meaning, in which, coherently with the dual process of mourning proposed by Stroebe and Schut [65], one can observe the oscillation between loss-oriented inclinations and the creation of new meanings. A balance between oscillations is known to favour better adaptation to the loss [66].

The analysis carried out in this study also made it possible to describe the links between the various themes identified (Figure 1). The feelings of the recipient are influenced by the ways in which the communication takes place; they are linked to the support (both sought and received), and modulated by the more or less sensitive methods implemented by the notifier while breaking the bad news. Finally, both feelings and support have an influence on the adaptation strategies implemented by the recipient. The lived experience of the recipient remains the central element in the death communication process. For this reason, it should represent the fulcrum to which notifiers should pay maximum attention during communication. Their behaviour and attitude can greatly mitigate the powerful feelings originating by such an impactful type of news. 

Given the exploratory nature of this study, several limitations should be noted. First of all, the recruitment of the sample was quite difficult, probably due to the peculiarity of the topic both in terms of numerical dimension (in many cases of suicide, it is the partner or family member to discover the body) and willingness of survivors to revisit a painful event. Therefore, the study did not account for sample representativeness, distribution with respect to gender and kind of death, and the generalizability and comparability of results. There is a small number of studies that target the recipients, and even smaller is the number of studies that specifically deal with the notification of unexpected and violent death. This imposes a cautious approach to the results of our study and prevents the formulation of too broad a conclusion. In addition, the online self-administered questionnaire narrowed the field of possible answers; in many cases, these were short and poorly articulated. Future research, therefore, should aim at person-to-person interviews, and a more balanced sample of participants either in terms of age, gender, time-distance from the event, type of death and also type of notifiers. 

With the materials collected in the present study, it would have been tempting to compare police officers versus health professionals in the role of violent death notifiers. However, the qualitative nature of the present study did not facilitate the comparison between the two main types of notifiers. Quite clearly, there might be relevant differences between those types of professional figures in relation to breaking bad news. Future research could consider structuring the sampling based also on the professional environment, frequency of the act of death notification, and educational background.

## 5. Conclusions

This study presented the experience of death notification as lived by family members who lost a loved one in unexpected and violent circumstances. The collected evidence shows that for survivors, in addition to the verbal content of the death notification, the non-verbal aspects, the means and the *lieu* in which the death was communicated also impacted their mourning experience. Negative experiences were related to notification taking place in a chaotic environment and being communicated by telephone or immediately guessed through the notifier’s gaze or tone of voice. Many participants reported the need to obtain formal support immediately after the communication.

The primary need of the recipients is therefore that the notification be performed in a sensitive and empathetic way. The notification should be provided in person, possibly by communicating clear and detailed information of what has happened in a private and quiet place, where survivors can freely express their emotions. These modalities could in fact mitigate the trauma of loss [11]. Many participants underlined the need for psychological support from the very first moments following the notification. After a traumatic death, coordination between the various professional figures would be a desirable requirement for the proper care of the bereaved person. 

## Figures and Tables

**Figure 1 ijerph-19-10709-f001:**
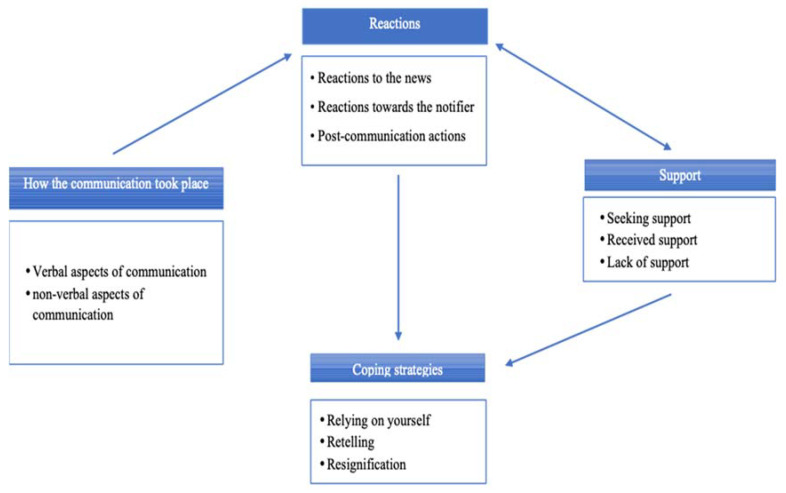
Roadmap of interactions among the main themes.

**Table 1 ijerph-19-10709-t001:** The ad hoc questionnaire.

*Questions related to demographics and the loss* 1. Age 2. Gender 3. Nationality 4. Family situation 5. Children 6. Educational title 7. Profession 8. What is your relationship with the lost person? 9. How long ago the loss happened? 10. Under what circumstances did your loved one pass away? *Questions about your experience of receiving notification of unexpected and violent death* 11. Did you personally receive the news of unexpected and violent death (suicide, homicide, traffic accident, work accident, natural disaster) of a loved one by professional figures? 12. Who communicated the news to you? 13. Where did the communication take place? 14. How was the news communicated to you? (In person, by telephone, by other means…) 15. Did you receive clear and accurate information about what happened? 16. Were there any particularly unpleasant aspects of the communication? If yes, please describe. 17. How do you evaluate the manner in which the communication took place? 18. How did you feel when you received the notification? 19. How did you react toward the person who gave you the news? 20. What struck with you the most about that moment? 21. What did you do after the communication? Was there anything that gave you support? 22. What would you have needed? 23. Would you like to add something? If so, please share your thoughts with us. Thank you!

**Table 2 ijerph-19-10709-t002:** Age, degree of kinship, time since death, circumstances of death, who made the communication, *lieu* where the communication took place, and medium used for communication.

	*N*	Percentage
*Age*		
21–30	9	17.3%
31–40	5	9.6%
41–50	11	21.2%
51–60	17	32.7%
61–70	10	19.2%
*Degree of kinship*		
Father/Mother	19	36.5%
Son/daughter	9	17.3%
Brother/Sister	12	23.1%
Husband/Wife	7	13.5%
Partner	1	1.9%
Daughter-in-law	2	3.8%
Friend	2	3.8%
*Time (in years) since death*		
41–31	2	3.8%
30–21	1	1.9%
20–11	5	9.6%
10–6	16	30.8%
5–1	24	46.2%
*Circumstances of death*		
Homicide	1	1.9%
Suicide	28	53.8%
Road accident	22	42.3%
Accident in the mountains	1	1.9%
*Who made the communication*		
Police officer	14	26.9%
Carabiniere	16	30.8%
Medical doctor	11	21.2%
Nurse	5	9.6%
Ambulance operator	2	3.8%
Doctor and nurse	4	7.7%
*Lieu where the communication took place*		
Law enforcement office	9	17.3%
House	22	42.3%
Hospital	14	26.9%
Train	1	1.9%
Car	1	1.9%
Ambulance	1	1.9%
Road	4	7.7%
*Medium used for communication*		
In person	34	65.4%
In person, after anticipatory call	7	13.5%
On the phone	11	21.2%

**Table 3 ijerph-19-10709-t003:** Themes, sub-themes, codes and frequencies observed in the sample.

	Sub-Themes	Codes	Frequencies
**Theme 1. How the communication took place**	Verbal aspects of communication	Clarity of presentation	26
		Generic information	21
	Non verbal aspects of communication	News intuition	8
		Notifier empathy	5
		Notifier vicinity	13
		Attentive and sensitive notifier	11
		Coldness of notifier	14
		Embarrassed notifier	8
		Unattentive and unpleasant notifier	4
**Theme 2. Reactions**	Reactions to the news	Shock	16
		Sense of emptiness	10
		Disbelief	16
		Estrangement	10
		Emotional Trauma	5
		Pain	35
		Despear	15
		Feeling of dying	2
		Death wishes	4
		Giving up	4
		Lack of the person	5
		Body reactions	6
		Guilt feelings	1
		Stiffening	5
		Indescribable emotions	8
	Reactions towards the notifier	Anger	10
		Attempts to deny	6
		Identification with the notifier	2
		Peacefulness	4
		Gratitude	8
		Detachment	7
		Silence	4
	Post-communication actions	Collaboration with notifier	10
		Body recognition	14
		Organ donation	8
		Farewell	12
		Return to the scene of the accident	3
		Return home	4
		Funeral organization	2
		Communication to friends and relatives	10
**Theme 3. Support**	Seeking support	Formal support	13
		Informal support	10
	Received support	Concret help	5
		Moral comfort	18
	Lack support	Practical	3
		Emotional	4
		Institutional	2
**Theme 4. Coping strategies**	Relying on yourself	Inner strength	2
		Family responsibility	4
	Retelling	Writing	5
		Partecipation to research on the topic	3
	Resignification	Reorganization of everyday life	5
		Search for explanations	3
		Loss as a learning	8

## Data Availability

The data presented in this study are available upon reasonable request to the corresponding author. The data are not publicly available due to their confidential nature.

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
