# Peer review of "Receiving Notification of Unexpected and Violent Death: A Qualitative Study of Italian Survivors"

_ijerph, 2022, doi:10.3390/ijerph191710709_

Round 1

Reviewer 1 Report

Overall: This study is an important study, which could impact how professional services deliver the news of death to families.  Considering the families experiences of death notifications, is an important angle to explore to find out how this impacts the start of their grieving process.  I believe the lack of data in this field to be a strength of this paper. 

There are two main concerns, the first is the first theme “how the communication took place”. This theme particularly “who made the communication”, “where the communication took place”, “medium used for communication” could be placed within the participant information table. This would allow for the other themes to be developed further and expanded upon. Which is where the second concern is, centred around the write up of the results section. This is written in a quantitative, narrative style. There is an over use of numbers to show that e.g. 10 out of the 50 participants had shown this.  

Specific Comments:

 Page 2 Line 8: “Compared to natural death, recovery from the experience” This reads as if you can recover from death. You are talking about mourning / bereavement here, but this is not clear.

Page 3 Line 72: “qualitative research design” – This needs more detail here of the online questionnaire.  Details of the questions asked in the online questionnaire are needed in the methods section.

Page 3 line 81: “ds” – is this standard deviation so SD.

Page 3 line 86: “year of death” It would be better to have time since death. As it is not stated when this research happened.

Page 4 line 92: It is stated you “was also conceived keeping in mind the danger of re-traumatisation of participants”. However, there is no detail to how this happened. Provide details of how this was done.

Page 4: Theme 1 How the communication took place. This is not a theme which helps to answer your aim of the study. This information is study relevant demographic information. I believe this would be better suited in the participant information table alongside age, gender etc.

Results section:  The results section is narratively written and is very quantitative based instead of qualitative based. Is there a rational for reporting how many participants reported x,y and z. I believe this takes away from the important of your data. This section should be rewritten to be more qualitative based.

Page 6 Line 154: It is stated that “Ten participants rated the death” this is not qualitative data. Rating would be quantitative.

Page 8 Line 237: “who lost a loved one in a car accident” this is an important aspect to report and should be included with more quotes to provide background to quotes.

Page 8 line 272: “likely to express their feelings of pain” You have stated more women reported reactions of pain. There needs to be a better link to the reactions theme and the literature stated here.

Page 9 line 309:  Godrum (2005) research, has this been shown within grief research, if yes this needs to be made clear.

Page 11 392: There is no figure to review.

Reviewer 2 Report

The study is interesting and the paper generally well structured. The following should be addressed in order for the paper to be in a more pusblishable shape.

1.      The study should be better justified. Authors should further expand the literature to show why this study is necessary. At the end of their introduction, the authors claim that there has not been another study in the past, which I found weird. If so, it would be interesting to know why this is the case. However, a quick search generated a few studies. In fact, it generated a systematic review by some of the authors. Lack of justification, as it stands, for the study is a major weakness.

2.      The authors should better explain why they chose snowball and not purposive sampling.

3.      The methodology is currently lacking. Is this a general inductive approach or thematic analysis? Why? Since the study has explored experiences, why it is not IPA (Interpretivist Phenomenological Analysis)?

4.      Have the authors checked for data saturation? If so, how and what did they find? If not, why not? Although saturation has been debated there are several approaches in the literature depending on the methodology (e.g., thematic saturation, meaning etc). In any case, the authors should explain how they know their data is enough to address their research aim.

  - Which guidelines for reporting qualitative research have the authors used? Is it COREQ, is it something else?

Round 2

Reviewer 2 Report

My comments have largely been addressed. The only thing is that the justification for the use of TCA should be included in the Methods.
